# Visualisation of Information Using Patient Journey Maps for a Mobile Health Application

Boram Lee [1], Juwan Lee [2], Yoonbin Cho [2], Yuan Shin [3], Chaesoo Oh [4], Hayun Park [5] and Hyun K. Kim [2,5,*]

1    Department of Industrial Design, Korea Advanced Institute of Science and Technology (KAIST), Daejeon 34141, Republic of Korea
2    School of Information Convergence, Kwangwoon University, Seoul 01897, Republic of Korea
3    The National Program for Excellence in SW, Kwangwoon University, Seoul 01897, Republic of Korea
4    Kakaohealthcare, Seongnam 13529, Republic of Korea
5    Department of Artificial Intelligence Application, Kwangwoon University, Seoul 01897, Republic of Korea
*    Correspondence: hyunkkim@kw.ac.kr; Tel.: +82-2-940-8143

**Abstract:** The demand for healthcare services using mobile devices has surged owing to the ageing population and increasing interest in personal healthcare. In particular, extensive efforts have been made to utilise mobile personal health records (PHRs) to provide personalised healthcare services to users (patients). Users must understand various types of health information that are included in PHRs to ensure successful and continued use of mobile PHRs. In this study, we developed and evaluated a user-friendly method for delivering health information from a PHR using mobile devices with small screens. We first constructed a patient journey map (PJM) for easy verification of disease treatment data from the perspective of the patient. Subsequently, we developed a mobile prototype that organises and visualises personal health information according to the patient-centred PJM and conducted user evaluations with 20 Korean participants. The results demonstrated that information delivery using the proposed prototype was easy to understand, user-friendly, and efficient. This paper highlights the importance of PJMs for patients in the understanding and use of different medical information. The proposed method is expected to promote the development of patient-centred mobile health applications in the future.

**Keywords:** personal health records visualisation; patient journey map; mobile healthcare app

## 1. Introduction

In recent years, the demand for mobile healthcare services has expanded significantly owing to the ageing population, increasing interest in personal healthcare, and popularisation of smart devices. Mobile healthcare refers to the use of mobile devices to provide healthcare services and information to individual users, whereby they can check their health status and access healthcare services [1,2]. Compared with traditional face-to-face services, mobile healthcare services are time- and cost-efficient and enable simultaneous and rapid distribution of information to numerous users.

Various mobile healthcare services are available in the market at present [3]. Mobile healthcare services can be broadly categorised into two types depending on the user: those for medical providers and those for non-medical users. Services for healthcare providers include electronic health records (EHRs), also known as electronic medical records (EMRs), which provide the health information of a patient (past medical history, medications, progress records, etc.) to medical staff and medical institutions [4–6]. Services for individual users include telemedicine, appointment management and reminders, self-diagnosis, health habit tracking, fitness and wellness tracking, mental health diagnosis, and personal health records (PHRs) [7]. In recent years, the development of mobile PHRs has gained traction to provide safer and more personal forms of healthcare to patients [7,8].

PHRs comprise electronic medical charts that contain medical data that are managed by a patient as well as information regarding the patient [9,10]; they include both personal healthcare services (that are provided based on personal health-related information) and platforms (that provide personal health information or personal healthcare services) [11]. Patients can remotely access information such as test results and prescriptions, add personal information and medical histories to their medical records, and check and manage their health statuses using mobile PHR applications [10,11]. PHR-related research in Korea has recently gained momentum; the Korean Government has created a PHR utilisation ecosystem as an important national agenda and announced plans to introduce means of improving national health and providing innovate medical services through personalised health information utilisation [12,13].

A PHR contains medical information and is specifically designed for users without or with limited eHealth literacy, which refers to the ability to access, discover, understand, evaluate, and electronically apply health information to make health-related decisions [14]. Therefore, the information contained in PHRs must be accessible and easy to understand for such users to promote the widespread utilisation of PHRs [15]. That is, an individual must be able to understand and access their health data to increase their access to healthcare, which is key to improving the quality of their PHR experience.

The health data that are provided to users through PHRs are divided into three types: medical institutional information (medical records, medication prescriptions, test results, consultation records, medical images, etc.), personal health information (pulse, blood sugar, diet, lifestyle, exercise, etc.), and public information (immunisation, weather, environmental records, epidemic information, etc.) [16]. Medical institutional information contains medical information, which is primarily managed by medical staff [6]. Therefore, ordinary users cannot understand and access such information, which suggests that user-friendly information delivery methods are required for users to access institutional medical information in PHRs.

Therefore, we propose a patient journey map (PJM)-based information delivery and visualisation method to help patients understand and utilise healthcare information using mobile devices. A PJM illustrates the interaction process between healthcare professionals and patients, with a focus on the touchpoints in the treatment journey, and provides an effective tool for visualising and understanding healthcare-related processes and information [17]. As a PJM is used to visualise the patient experience, it can be used as a tool by both healthcare professionals and patients. With the aid of PJMs, patients can understand the treatment process, access information on their disease, and utilise healthcare services [18]. However, existing PJMs are exclusively designed for healthcare professionals to aid them in understanding how patients use hospitals or health systems and to improve the quality of healthcare services by facilitating smooth communication between healthcare parties. Therefore, the direct application of existing PJMs to patient-centric services is a considerable challenge.

In this study, we developed a PJM with patients as the primary users to manage personal medical data and check health status. We used this PJM to construct and evaluate mobile information delivery and visualisation methods. The study was conducted with 20 Korean subjects.

## 2. Review of Literature and Applications

### 2.1. PJMs

A PJM depicts the interactions between healthcare professionals and patients around touchpoints during a patient's treatment journey. To date, PJMs have been primarily used by healthcare providers to communicate and share information in the treatment patients. In recent years, healthcare has become more patient-centred, and PJMs have also been adapted accordingly and are now used to provide high-quality care to patients while also engaging patients and their caregivers in care tasks and significantly influencing their care.

In the present study, we initially identified the characteristics and components of PJMs proposed in previous studies to define a patient-centred PJM. Percival et al. [19] proposed and evaluated a novel patient journey modelling technique, PaJMA, to improve patient care by addressing the challenges of integrating clinical processes and information systems with traditional clinical practice. Specifically, they proposed a patient journey model that provides a visual representation of processes, technologies, and stakeholder interactions, which are used to integrate clinical processes, information systems, and health information technology (HIT) systematically to support a patient. Their aim was to integrate clinical processes with information systems and HIT to benefit patients and staff and thus aid in healthcare delivery.

McCarthy et al. [20] introduced a design tool called integrated patient journey mapping (IPJM) to help multidisciplinary teams design effective HIT solutions by considering three key elements of healthcare quality: clinical effectiveness, patient safety, and patient experience. To support requirement analysis and foster empathy and understanding among multidisciplinary teams, IPJM uses 'patient personas, medical timelines, and medical pathways' to construct a picture of a patient's movement and task performance along a medical pathway.

Borycki et al. [21] integrated various technological tools into PJMs to improve healthcare quality. Their ultimate goal was to improve the quality of healthcare by providing support for continuous medical care, reducing wait times, and enhancing patient safety. They utilised PJMs as tools to indicate the touchpoints where new technologies and tools should be introduced to improve the healthcare journey of a patient and create a digital health strategy.

Agarwal et al. [22] used PJMs to address three breakdowns (communication, patient learning, and system breakdowns) that can affect care transitions (i.e., movement of patients between multiple healthcare systems). These authors suggested the application of PJMs to coordinate and align multidisciplinary care teams for patients undergoing care transitions. Their objective was to improve the quality of care by identifying problems in care transitions from the patient's perspective rather than from the traditional healthcare provider's perspective, which is solely governed by quantitative efficiency.

Although PJMs are extensively used in healthcare, they are not well documented. Simonse et al. [23] reportedly constructed PJMs that include the patient perspective to innovate healthcare delivery. They proposed a PJM design method based on the advantages of service design; this PJM design method can identify problems in current services through user-centric experience management.

O'Dell [24] performed an applied case study using PJMs and proposed a service design framework that embraces new technologies to improve patient experience. The author interviewed patients to establish disease journeys and generated a PJM based on various disease cases. The author reported that the generated PJM can serve as a communication tool for designers when identifying challenges in a healthcare system and the opportunities for integrating technology to improve user experience.

The aforementioned studies were focused on exploring the potential applications and utilisation of the PJM method and indicated the effectiveness of PJMs in understanding patient experiences and facilitating communication between stakeholders. Further, in these studies, 'Main Timeline' and 'Main User' were used as the PJM components. Main Timeline is an essential element required to describe a patient's treatment. Main User refers to the actual user of the PJM, and, except for the study by Borycki et al. [21], other studies set medical personnel as the actual user. This is because the existing PJMs are mainly used to help medical staff and related parties in the medical field in decision making, and it is difficult to deliver information to patients according to the existing PJMs. Therefore, establishing a patient-centric PJM as a communication tool between patients and the healthcare system is necessary to make patients more aware of their disease and help them in actively participating in the treatment process.

### 2.2. Mobile Health Applications

Mobile healthcare applications (apps) can be broadly divided into two categories based on their target audience: those for healthcare providers and those for the general public, including patients [25]. Apps for healthcare providers are primarily developed for healthcare workers who provide healthcare services to patients. These apps facilitate EHR/EMR access, enable remote patient monitoring and telemedicine, allow for medication specifications, and aid healthcare professionals in providing recommendations for diagnosis and treatment. Apps for patients and other individuals include individual care apps (e.g., fitness, sports, games, and auto-diagnosis, which allow users to manage only the data that are generated directly by the user, such as steps, pulse, blood pressure, and sleep patterns), apps to check their PHRs, apps to contact their healthcare professionals, educational health apps, and social networking apps [25].

In this study, we propose a visualisation approach for analysing medical information in personal health management. To this end, we examined the medical information representation of seven mobile health apps that are used by patients and individuals in Korea: three hospital apps that were developed by large hospitals and four health management apps that require long-term recording of medical data, such as diet and blood pressure. We selected these apps with the aim of exploring how health information is presented depending on its source, i.e., health information provided directly by healthcare organisations and health information created and managed by individuals.

The three hospital apps provide easy access to patient health information; they primarily import data from the hospital database, without the need for patients to create their own medical records and information. These apps are generally designed to facilitate smooth and efficient hospital care and provide features such as patient appointment scheduling, searching for doctors, scheduling and viewing of appointments, and viewing of personal patient information. In contrast, the four health management apps share information on specific diseases among users, such as diabetes and cancer, provide treatment directions for diseases according to the health information of an individual, and synthesise diet and exercise records in the form of reports. In these apps, the user must manually enter various types of health information, including their treatment stage, which may be inaccurate because the available information is limited and relies on personal input. The features that are provided by these apps include directional information on the treatment stages of various diseases, questions that are frequently asked by patients and their answers, exercise records, blood sugar levels, dietary recommendations, and personal health profiles.

The aforementioned healthcare apps provide information on medical treatment and health records that are accumulated over time in an enumerated format, with time as the main axis (the information is not categorised by disease type). In particular, as the primary feature of a hospital app is the scheduling of medical appointments, information on these appointments (medical staff details, test items, office location, etc.) is presented according to a schedule. These healthcare apps display either the health records that are entered by the user over time on a calendar or the current stage of the disease treatment of the patient, with simple information mapped to that stage. However, this approach enumerates information that is collected over time; as a result, the user receives fragmented information. Moreover, most apps are dependent on specific medical institutions or healthcare devices. Therefore, only limited information is available, and the integration of PHRs and their presentation according to certain standards becomes arduous.

However, with the accelerated development of platforms for PHR-related services in recent years, various technologies have been established for viewing the medical data of individuals without depending on specific medical organisations or services. The next step involves the integration of the scattered medical information of individuals and development of a representation method that will enable individuals to view and access information with ease.

### 2.3. User Experience Evaluation Metrics for Mobile Health Apps

We propose a novel visualisation method for effective and user-friendly information presentation using mobile health apps. The results of user experience (UX) evaluations are presented and discussed in detail in this paper. Evaluation metrics are required to assess the UX with mobile health apps for user evaluation of the proposed prototype. In this section, we introduce various existing metrics for evaluating mobile healthcare apps and describe the metrics that are used in this study.

Traditionally, common ways to measure the UX of an interactive product include the AttrakDiff questionnaire, the User Experience Questionnaire (UEQ), or the modular evaluation of key Components of User Experience (meCUE) questionnaire [26–29]. AttrakDiff assesses the user's feelings about the cognitive artifact with the help of 28 pairs of opposite adjectives [26]. Users can evaluate both the perceived pragmatic quality, the hedonic quality, and the attractiveness. The UEQ covers the following user experience aspects: attractiveness, efficiency, perspicuity, dependability, stimulation, and novelty [27]. It allows the users to express feelings, impressions, and attitudes that arise when experiencing the product under investigation in a very simple and immediate way. meCUE was developed to measure key components of UX in a comprehensive and unified way. meCUE is a questionnaire with 34 items covering the following components: product perceptions (usefulness, usability, visual aesthetics, status, commitment), user emotions (positive, negative), consequences of use (intention to use, product loyalty), and overall evaluation [28]. However, these methods are not specialised for mobile healthcare services and are instead aimed at general interactive systems.

Various survey methods have been used to evaluate the user experience of mobile healthcare services. These questionnaires were primarily used to assess usability, quality, acceptance, and satisfaction. The System Usability Scale (SUS), mobile application rating scale (MARS), and Post-Study System Usability Questionnaire (PSSUQ) are commonly used for mobile healthcare application usability testing [30].

The System Usability Scale (SUS) was developed to provide a quick and simple method for measuring the usability of any system [31]. The Post-Study System Usability Questionnaire (PSSUQ) measures user satisfaction with system usability and is a usability evaluation tool specifically developed for use in the context of scenario-based usability testing [32,33]. These two questionnaires are commonly used to evaluate the usability of various systems because of their reproducibility, reliability, and validity. However, they are not specifically designed to evaluate the user experience of mobile healthcare apps and it is difficult to evaluate them considering the characteristics of mobile healthcare apps.

The mobile app rating scale (MARS) was used to evaluate mobile app quality. It is designed to be broader than its basic usability. MARS has moderately validated psychometric properties and is widely used to measure the quality factors of numerous mobile healthcare applications [34]. However, the target audience of the MARS is experts in a field (researchers, clinicians, and other professionals) who want to evaluate mobile healthcare apps, and raters need to learn enough to utilise this evaluation tool. To compensate for this, the authors developed uMARS, a simple user version of MARS targeted at general users for usability evaluation by nonspecialised end users. Although MARS and uMARS include usability components such as immersion, functionality, and information quality, as well as subjective quality factors of apps, these factors have not been validated, making them unsafe evaluation tools.

The Health-ITUES is a customised tool developed to assess the usability of healthcare mobile technologies with reference to various system usability evaluation questionnaires, such as the IBM Computer System Usability Questionnaire, Technology Acceptance Model (TAM), Unified Theory of Acceptance and Use of Technology (UTAUT), and Questionnaire for User Interaction Satisfaction (QUIS) [35]. The authors demonstrated the reliability and validity of the questionnaire in a mobile healthcare application usability study. However, unlike existing measurement tools, the Health-ITUES was designed to be customised through changes to several questions in the questionnaire so that the user–system–task–

environment interactions of the app to be evaluated can be considered [36]. This can reduce the reliability of the evaluation because subject responses may vary depending on the question. In addition, the Health-ITUES is a validated evaluation metric centred on perceived usefulness and perceived ease-of-use factors, but with a focus on usability.

The mHealth App Usability Questionnaire (MAUQ) is a questionnaire designed to evaluate the usability of mobile healthcare apps, which was developed based on well-validated questionnaires from previous mobile app usability studies [37]. Furthermore, the questionnaire was designed considering the problems of the mobile application rating scale (MARS) and Health Information Technology Usability Evaluation Scale (Health-ITUES). Although the MAUQ is reliable and valid for assessing mobile healthcare app usability, it lacks factors and subscales that consider health literacy, which is an important concept in mobile healthcare apps. One study recommended the inclusion of health literacy-related assessments in the usability testing of patient web portals to improve the adoption of consumer health information systems and the health knowledge of users [38]. This finding suggests that information quality in health information systems should be assessed in multiple ways. Five dimensions of information quality have been identified for health websites: accuracy, completeness, depth, understandability, and relevance of the information [38]. Among these, understanding of information is rated as one of the most important qualitative dimensions [38]. Information understandability refers to the readability of information in plain language, including textual statistics, explanations of medical language and acronyms, selection of a display format for numerical or graphical information, and image clarity.

Most evaluation factors for healthcare apps have been developed in terms of usability and quality [30,39]. These two factors are commonly included in user satisfaction evaluation tools and are centred on usability. Therefore, existing mobile healthcare app evaluation tools lack questions on factors such as information literacy in relation to personal health data communication, which makes it difficult to comprehensively measure the experience of a patient with mobile healthcare apps.

To overcome this issue, Kim et al. [40] developed a UX evaluation scale for mobile healthcare apps by collecting evaluation items for each factor from existing related evaluation methods (MAUQ, uMARS, Health-ITUES, and eHLQ), combining items with overlapping or ambiguous meanings, and analysing their validity through factor analysis. In general, the UX consists of usability, emotion, and user value [41,42]. The UX evaluation scale offers the advantage of considering aesthetics and health literacy in addition to usability and usefulness, which are the focus of existing evaluation methods; it can be compiled through a simple questionnaire. Therefore, this UX evaluation scale was adopted to measure the user experience of the prototype, and the detailed survey questions are presented in Table 1.

**Table 1.** User experience evaluation scale of mHealth apps.

| Factor | Questions |
| --- | --- |
| **Ease of Use and Satisfaction** | Q1. The app was easy to use. |
| | Q2. The interface of the app allowed me to use all the functions offered by the app. |
| | Q3. I feel comfortable using this app in social settings. |
| | Q4. The amount of time involved in using this app has been fitting for me. |
| | Q5. Overall, I am satisfied with this app. |
| **Information Architecture** | Q6. Whenever I made a mistake using the app, I could recover easily and quickly. |
| | Q7. The navigation was consistent when moving between screens. |

**Table 1.** *Cont.*

| Factor | Questions |
|---|---|
| Usefulness | Q8. This app has all the functions and capabilities I expect it to have. |
| | Q9. The app would be useful for my health. |
| | Q10. The app helped me manage my health effectively. |
| | Q11. This mobile health app provided a suitable way to get medical care. |
| | Q12. The app improved my access to health care services. |
| Easy Understanding of Information | Q13. The medical/health information in the app is accurate, well written, and relevant to the purpose of the app. |
| | Q14. The information provided by the app is comprehensive and concise. |
| | Q15. Descriptions of visual information (charts, graphs, images, etc.) provided by the app are logical and clear. |
| | Q16. The information provided by the app was easy to understand. |
| Aesthetics | Q17. This app is using the appropriate colours. |
| | Q18. I like the menu structure and design of the app and find it easy to use. |

| Answers (7-point Likert scales) | | | | | | |
|---|---|---|---|---|---|---|
| **1**<br>Strongly disagree | **2**<br>Disagree | **3**<br>Somewhat disagree | **4**<br>Neutral | **5**<br>Somewhat agree | **6**<br>Agree | **7**<br>Strongly agree |

## 3. Research Process

This study consists of three main phases: development of a patient-centred PJM, development of a UX prototype to effectively deliver PJMs in a mobile environment, and UX evaluation of the prototype (Figure 1). The persona technique was used to develop a patient-centred PJM and a prototype. The patient-centred PJM was developed through consultation with medical staff after organising the disease journey according to the common components of a PJM derived from the existing literature and the personas established in this study. A pilot prototype was developed using the proposed PJM. The final prototype was designed based on insights gained from the pilot study and design guidelines derived from a literature review on health information visualisation. Finally, we conducted a user evaluation of the proposed prototype. The user evaluation consisted of a survey, observations, and in-depth interviews.

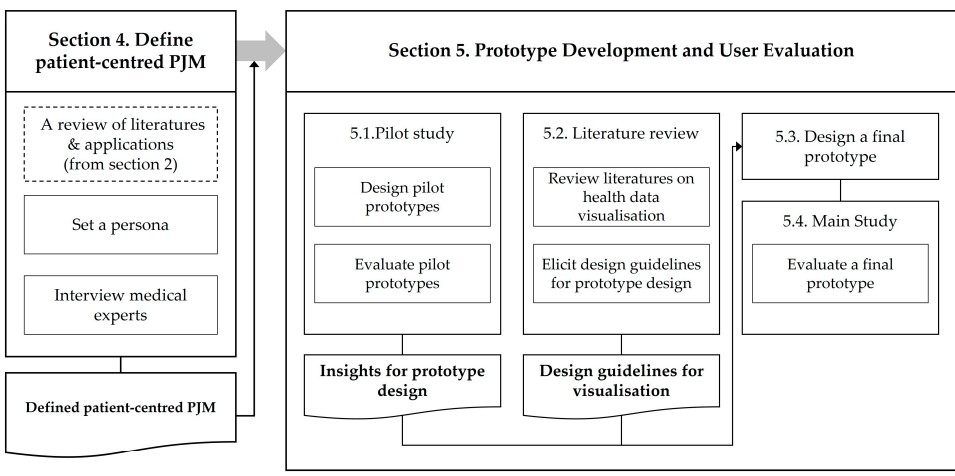

**Figure 1.** Research process.

## 4. Defining Patient-Centred PJMs

### 4.1. Methods

The development of the patient-centred patient journey phases proceeded as follows. First, we established a persona for disease journey development, followed by prototype development and evaluation experiments. When a user test is conducted from a medical perspective, it is difficult to recruit users with the same disease; thus, the vignette technique, which is similar to the persona technique, is widely applied [43–45]. Two subjects in their 40s who suffered from chronic diseases were interviewed, and the persona was drawn up after asking about their current disease and health status. Subsequently, we developed a draft disease journey based on the persona. The PJM components from previous studies were used as references. Thereafter, we consulted four medical experts (dermatologist, psychiatrist, radiation oncologist, and internal medicine specialist) to determine the suitability of the disease situations and disease stages of the personas for real-world occurrences whether or not the patient journey stages needed to be revised or supplemented. Finally, we constructed the final patient journey stages and personas through revision.

### 4.2. Persona

The persona is a 47-year-old woman named 'Eunji Kim', whose detailed situation is shown in the Table 2. The age range was set to 40s, as she had relatively more experience with medical treatment and hospitalisation than those in their 20s and 30s, and the specifics are that she usually leads a healthy lifestyle. However, this year, she went on a trip to Jeju Island and suffered a neck disc herniation and fracture in a car accident during the trip. The four diseases she has suffered so far are set as diseases that can be encountered in daily life, and the diseases currently under treatment are high blood pressure and fracture, after which the disk and flu are cured. In the case of the fracture, it was being managed after treatment, but it fractured again and is currently under treatment.

**Table 2.** Persona description.

| Name | Age | Marital Status | Job |
|---|---|---|---|
| Eunji, Kim | 47 | Married | English Teacher |

| | Personalspecifics | | |
|---|---|---|---|
| **Daily life** | Thanks to her normal healthy lifestyle, Eunji Kim was only being treated for minor diseases without major diseases. She was diagnosed with hypertension in a medical examination in 2019 and began to take an interest in disease management. In the meantime, she suffered a neck disc herniation and a fracture due to a traffic accident on a trip to Jeju Island that year. The neck disc herniation is on the road to recovery through physiotherapy after surgery, and for the fracture, she is undergoing a new cast treatment as the bone that had been broken during medical rehabilitation broke again. She did not have to worry about her health before, but she is confused because she has to manage all kinds of diseases at once due to hypertension, neck disc herniation, and fracture. | | |
| **Health state** | **Diseases** | Under treatment: hypertension, fracture (relapse)<br>After treatment: neck disc herniation<br>Full recovery: flu | |
| | **Prescription information** | Hypertension: after hospitalisation, surgery was performed, medication was administered, and the symptoms were under control.<br>Fracture (relapse): the fracture that was under management has returned, and medication and cast therapy are being prescribed.<br>Neck disc herniation: recovery is under way through postoperative rehabilitation.<br>Flu: flu without vaccination, easily cured with medication. | |
| **Trouble** | She is confused by the sudden increase in the number of diseases she has to manage, so she is worried about how to understand and manage her illness. | | |
| **Core values and needs** | She wants to take care of her health while considering the diseases she has experienced so far. | | |

*4.3. Patient-Centred PJM*

The PJM has different components depending on its purpose and focus. However, certain components were present in the majority of the PJMs. The components that we identified as common across the six PJM-related studies reviewed in the first half of this thesis were the main timeline and users. The main timeline was an essential component to describe the patient's care, with three papers defining PJMs based on time and three papers defining PJMs based on the patient's stage of disease treatment. The main user refers to the actual user of the currently commercialised PJM, and, except for one paper, five papers set the actual user as medical personnel. This is because existing PJMs are mainly used to help medical professionals in the medical community make decisions and often consist of elements that are difficult or unnecessary for patients to understand. Therefore, patient-centred PJMs are required to ensure that patients are aware of their disease and can actively participate in the treatment process.

Therefore, the main timeline of the PJM in this study was organised into common treatment stages of the disease so that users could identify their treatment stage and provide medical information for each stage (Figure 2). Specifically, the disease journey stages were organised into onset, before treatment, during treatment, after treatment, and at full recovery. Onset is when you organise the events that led to your illness or the symptoms that made you realise you had an illness, before treatment is when you get medical care and tests to find the disease before it is identified, during treatment is all the care that doctors and other healthcare providers provide to you in the hospital, and after treatment is when you have completed the treatment provided by the hospital but are still recovering and caring for your body. Finally, full recovery is achieved when you recover to an approximation of your pre-illness health status. Based on these five stages, we received information on many detailed treatment tasks for personas from our medical advisors and sequenced them accordingly.

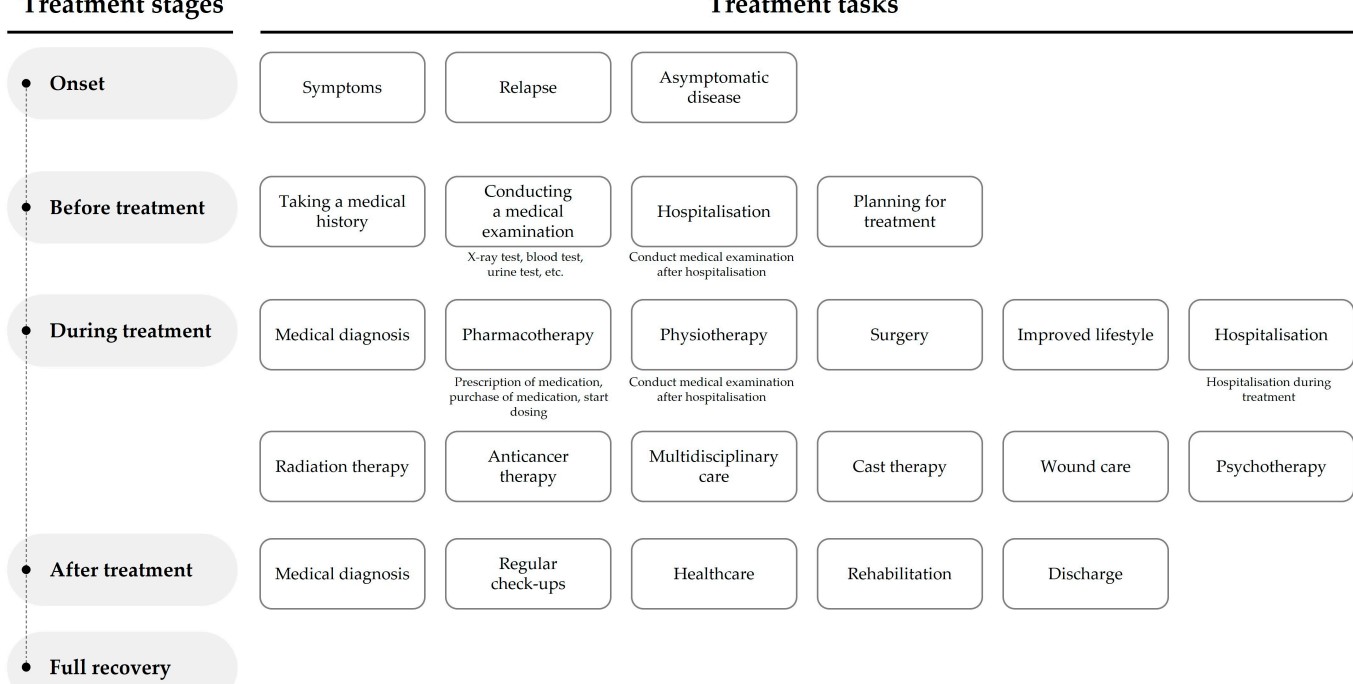

**Figure 2.** Patient-centred PJM.

## 5. Prototype Development and User Evaluation

### 5.1. Pilot Study

A pilot study was conducted prior to the development of the prototype based on a visualisation approach for effectively communicating the established patient-centred PJM to users. The aim of the pilot study was to explore the initial concept of the prototype and gain insights into its development. Three pilot prototypes were developed for the pilot study on which the user evaluations were conducted.

#### 5.1.1. Participants

For the pilot user evaluation, we recruited 16 participants in their 20s. Their average age was 23 years (SD = 1), and there were 13 men and 3 women. The participants were recruited based on their general interest in healthcare.

#### 5.1.2. Prototype Design

The prototypes for the pilot study were created through an ideation and design process based on a survey of existing PJM visualisations and included three different types (Figure 3). Prototype A is a vertical type that vertically expands the information scope when the user presses a specific button. Prototype B is a horizontal type that horizontally expands the information scope. Prototype C is a U-shaped type with a fixed U-shaped information scope.

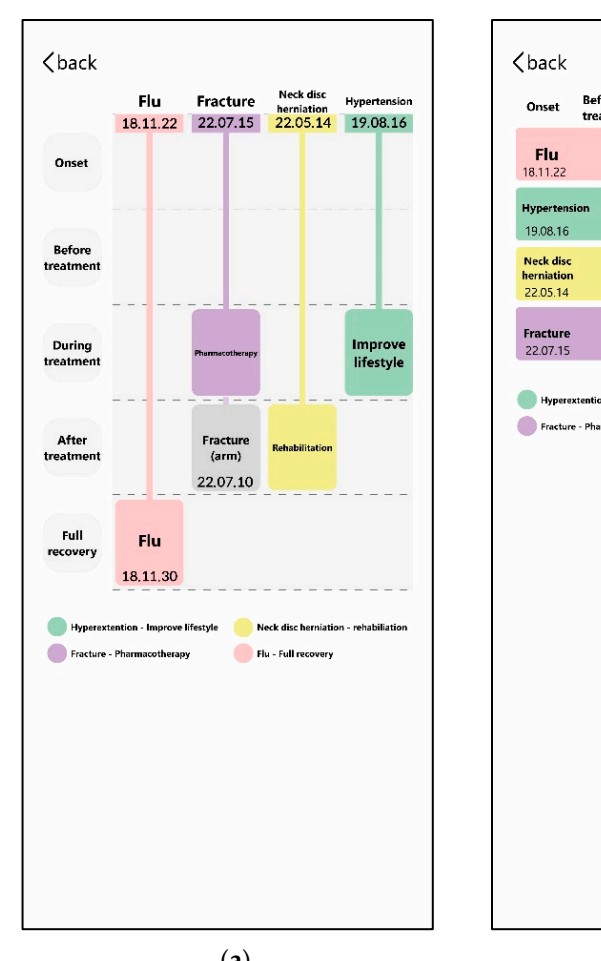
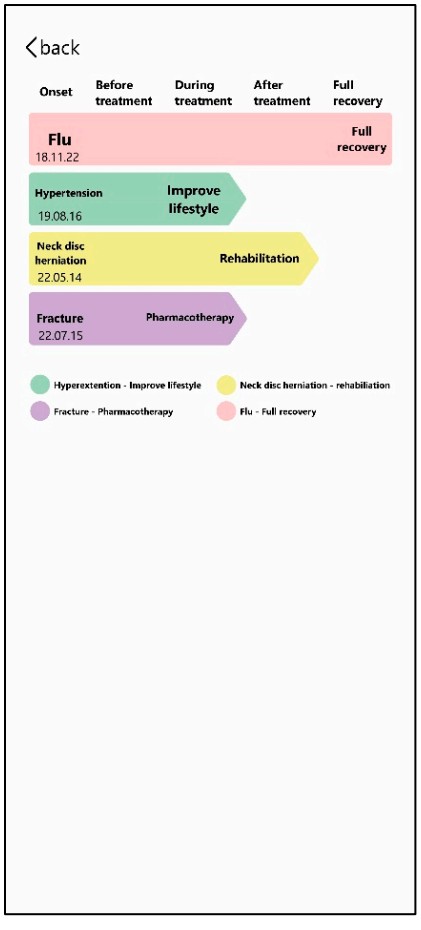
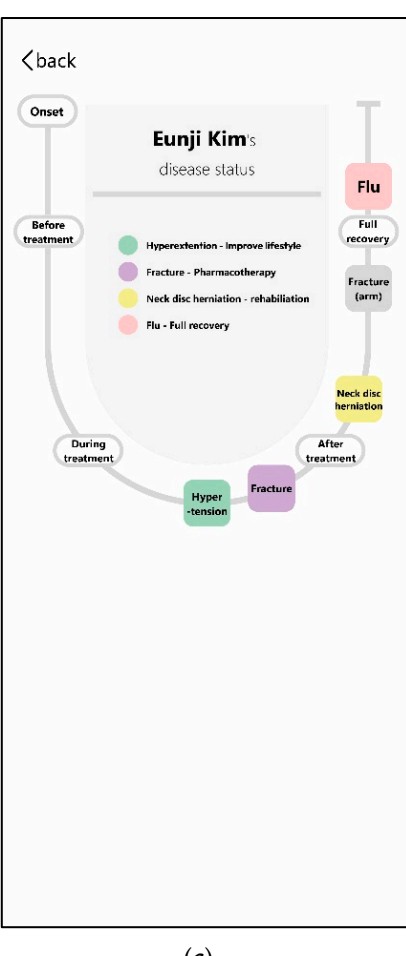

(**a**)  (**b**)  (**c**)

**Figure 3.** Examples of pilot prototypes. (**a**) Vertical type; (**b**) horizontal type; (**c**) U-shape type.

### 5.1.3. Methods

Following the pilot prototype design, we conducted a user evaluation assessment to determine the response of the users to the information delivery method that was presented in the prototype. During the user evaluation phase, participants identified their overall persona (Table 2) and patient-centred PJM (Figure 2). Subsequently, they responded to preset questions while freely using the prototypes that were developed for smartphones (Galaxy Note 20). The preset questions were designed to check the details of a disease considering all diseases and the relevant content listed in the prototype. The prototypes were used in a random order and questions were randomised based on disease type to cover the essential range of information for the participants during their treatment. Examples of the preset questions used in the prototype for extracting information are listed in Table 3. We also observed the pattern of the participants' use of the prototype. After performing all of the tasks, we quantified the user experience using the mHealth app user experience evaluation scale (Table 1), observed the participants (error measurements, behavioural observations, etc.), and collected detailed data on the user experiences through post-interviews, including information regarding advantages, disadvantages, and improvements that were required in the visualisation method. Thereafter, we analysed the collected observation, interview, and survey data.

**Table 3.** Examples of preset questions for extracting information from the prototype.

| # | Questions |
|---|---|
| 1 | Which of the medicines you take to treat high blood pressure is listed first? |
| 2 | How high did your fever rise when you got the flu? |
| 3 | As a result of the disc test, how many times did bones grow in the cervical spine? |
| 4 | When you saw the doctor for the fracture, which hospital did you see? |
| 5 | When did you start rehabilitation treatment for the disc herniation? |

The survey was divided into five categories, and the detailed questions for each category are listed in Table 1. The Ease of Use and Satisfaction categories consisted of five questions that assessed the difficulty of use, time required, and overall satisfaction with the visualisation method. The Information Architecture category consisted of two questions that assessed the information structure: one concerning recovery from mistakes and the other concerning the consistency of the information structure. The Usefulness category comprised five questions that identified the essential functions and degree of healthcare support. The Easy Understanding of Information category included questions that evaluated the difficulty and logic of the information provided by the visualisation. Finally, the Aesthetics category consisted of two questions that assessed the aesthetics and colours of the visualisation's buttons and components.

### 5.1.4. Results

According to the survey results, Type A scored the highest in the Ease of Use, Satisfaction, Usefulness, and Aesthetics categories, whereas Type B scored the highest in the Information Architecture and Easy Understanding of Information categories (Figure 4). The average score for all questions was the highest for Type A. This result was also confirmed by the in-depth interview responses, which revealed that the horizontal and vertical layouts were more comfortable to use and easier to navigate. However, these results were not statistically significant because there were only a small number of participants, and the scores for each prototype were slightly different.

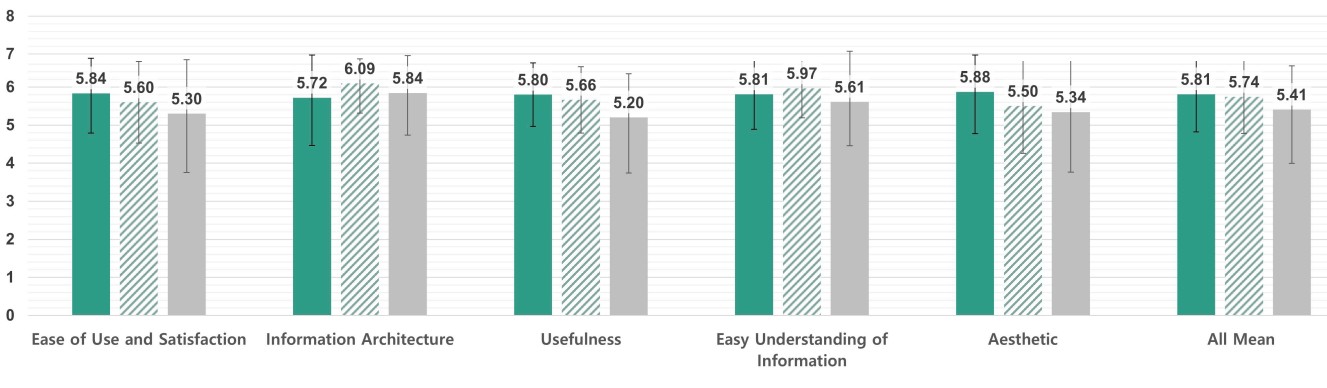

**Figure 4.** Survey results.

The advantage of the vertical and horizontal types is that they can provide more information in a large space. The results suggest that the process of using the vertical and horizontal layouts to navigate information on mobile interfaces is similar, which may have contributed to the familiarity of the participants with Type A and B and their favourable ratings. Prior to the experiment, the excessive amount of information and inconvenience of long scrolling, owing to the expanded information area, appeared as the major shortcomings of the prototypes. However, easy information selection is more important than a large information display area. Furthermore, the majority of participants did not experience scrolling-related issues, indicating that expanding the information area is more effective than providing information in a fixed area.

Based on the results of the pilot study, we decided to upgrade the vertical and horizontal visualisations that were evaluated during the pilot prototype study. Moreover, we investigated the placement of information in the final prototype design so that it could be easily navigated by considering scrolling and the amount of information, rather than attempting to reduce the information (owing to concerns regarding the amount of information).

*5.2. Literature Reviews for the Visualisation of Mobile Health Data*

We reviewed previous studies to derive guidelines for health data visualisation and incorporated them into the design of the prototype. Related studies for the literature review were collected by combining the keywords "health data", "healthcare", "visualization", and "visualization design" on Google Scholar, and a total of 20 papers were searched. Among the 20 papers, 10 major papers were finally selected based on the question "Have user-centred visualisation design considerations been presented?" Through this process, 10 papers that simply proceeded with visualisation were excluded. In this manner, visualisation guidelines were derived to develop a prototype for effectively visualising the disease journey stages, as outlined in Table 4.

Ledesma et al. [46] implemented and evaluated time-based INCLude visualisation software using an insight-based methodology. Their findings revealed that time-based data visualisation tools are more effective for understanding patient data. Islam et al. [47] conducted a study to understand smartwatch representations and suggested new visualisation methods. They surveyed 237 watch users and demonstrated that health data visualisation on smartwatches requires the active use of icons to understand medical information. Blascheck et al. [48] evaluated the speed at which users can perform simple data comparison tasks for small visualisations on smartwatches. They used bar, doughnut, and pie charts to perform a comparison task and showed that bar charts are more efficient for quick data comparisons. Islam et al. [47] proposed and evaluated different visualisations to understand the preferences and effectiveness of sleep data visualisation on smartwatches. They showed that users are interested in detailed sleep data and at-a-glance visualisations. Ola and Sedig [49] presented four visualisations to demonstrate visualisation design using

a framework-based approach. They showed that in time-based visualisations, the user should be able to identify the point of view.

Senathirajah et al. [50] described a methodological approach for addressing display partitioning in visualisation. They showed that display partitioning should be avoided owing to its disadvantages, which include complex navigation requirements and increased cognitive load, and that the data visualisation should be viewable at a glance on a single screen. Bastardo et al. [51] proposed a radial timeline model to provide a holistic view of a patient's medical history. To address display fragmentation, they implemented a visualisation that presented the entire patient history on a single screen without information loss and showed that the patient's condition can be rapidly assessed using appropriate colour codes according to the visualisation. Hossain et al. [52] investigated various medical history visualisation tools to support doctors and proposed a personal health data visualisation tool that visualises Gantt charts. They reported that the visualisation of health records should be accurate as well as easily and quickly understandable. Arcia et al. [53] investigated user preferences and perceived meaning among infographics in health data visualisation and suggested that visualisations should be information-rich, provide context, and use friendly colours. Rajabiyazdi et al. [54] designed a technical solution to address the problem of presenting, reviewing, and analysing patient-generated data. They demonstrated that the visualisation's design should include contextual data, allow users to define the amount of information displayed, and offer at-a-glance features.

Based on the above-mentioned studies, we developed visualisation guidelines for health data (Table 4) and designed a prototype using these guidelines.

**Table 4.** Design guidelines for health data visualisation.

| Design Guideline | References |
|---|---|
| In health data visualisation, time-based visualisations help you understand your data better than non-time-based visualisations. | [46] |
| Medical information is unfamiliar data to the ordinary user, so you should actively use icons for better understanding. | [47] |
| Bar graphs are a great way to present data that need to be quickly viewed and understood. | [47,48] |
| At-a-glance data visualisation. | [47] |
| Visualisations should contain detailed and accurate data. | [48,51–53] |
| In time-based data visualisations, users should be able to identify the point in time (year, month, date, hour, etc.). | [49] |
| Data visualisations need to be within a single screen. | [47,50,51,54] |
| Using proper colour coding can improve the understanding of your visualisations. | [51,53] |
| The visualisation should be easy and quick to understand. | [52] |
| The user should be able to choose how much information is displayed. | [54] |
| The visualisation's design should include information about context. | [53,54] |

### 5.3. Final Prototype Design

The final prototype was developed through an iterative design process based on the results of the pilot study and the derived set of design guidelines (Figure 5). The vertical and horizontal information arrangements that were highly rated in the pilot study were used as a starting point for the design and application.

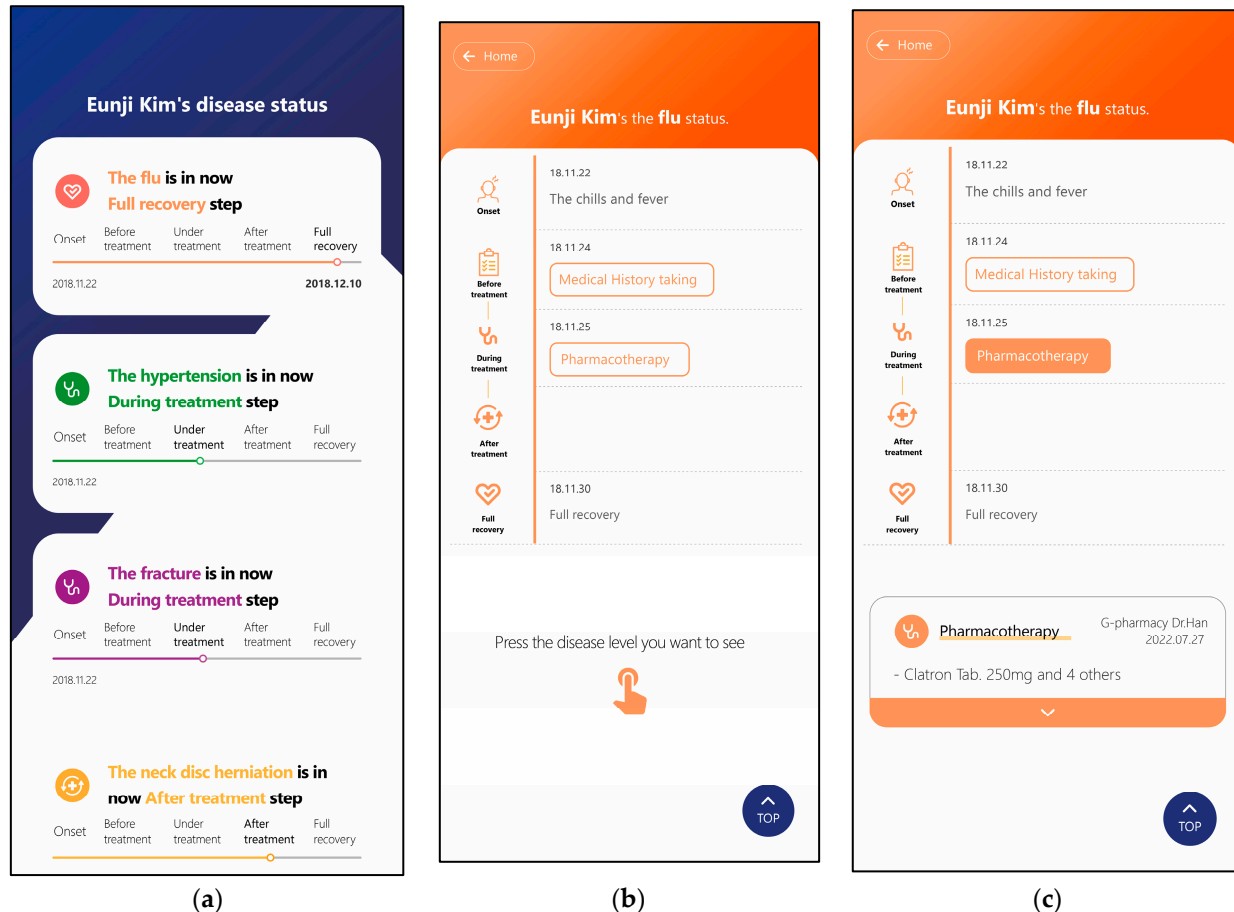

**Figure 5.** Final prototype: (**a**) home screen; (**b**) detailed screen; (**c**) detailed task/information selection screen.

The final prototype was a card-based home screen that displays the stage of each disease at a glance. The home screen was designed by applying the concept of horizontal visualisation, which shows the treatment status of a disease horizontally and separates each disease unit into cards. Cards are characterised by their ability to show the stages of a disease in a smaller area than other visualisation formats. The cards on the home screen represented each disease and each card indicated the progress of treatment according to the PJM's stage of disease as a bar graph. The text (e.g., "Flu is now in Full recovery step") explained the current stage of disease treatment so that the disease situation could be understood at a glance.

Moreover, the main colour theme of the card differed for each disease, making it easier for users to distinguish between diseases. Icons representing the treatment steps were used to help users understand the steps more intuitively.

Each disease card could be tapped to enter a detailed screen that displayed treatment stage information for each disease. This detailed screen was established in a vertical format so that the user could view the detailed stages of the disease from top to bottom. The treatment stages displayed relevant information, including the treatment tasks for each stage, which were set as buttons so that when the user pressed the button, the relevant information appeared at the bottom of the screen. This prevented users from scrolling through unnecessary information and allowed them to view the necessary data selectively.

### 5.4. Main Study

This study was conducted with the approval of the IRB (7001546-202300331-HR(SB)-003-02) of Kwangwoon University.

### 5.4.1. Participants

User evaluation was conducted identically to the user evaluation phase of the pilot study. We selected 20 participants: 10 in their 20s (mean = 23.5 (SD = 1.7); male: 7; female: 3) and 10 in their 40s to 60s (mean = 53.8 (SD = 7.1); male: 3; female: 7). The participants were selected based on their use of health-related apps and their general interest in health.

### 5.4.2. Methods

The user evaluation of the main study was conducted in the same manner as the pilot study described in Section 5.1.3. First, after going through the process of immigrating to the persona, the participants attempted to use the final prototype, as shown in Figure 5. A survey was conducted using the mHealth app's UX evaluation scale (Table 1). Subsequently, through in-depth interviews, insights into the pros and cons of the visualisation, necessary functions, and visualisation information were sought. The interview contents were transcribed verbatim and analysed along with the observational data.

### 5.4.3. Results of the Main Study

**Daily healthcare:** Participants in their 20s were the most likely to manage their health through exercise, diet, and nutritional supplements, with three out of ten using a mobile health app (i.e., an exercise tracker). These participants tended to focus more on managing their health habits than on managing their diseases. Among the participants in their 40s to 60s, five out of seven were taking care of their health, including regular health check-ups, the use of nutritional supplements, exercise, and diet. Two out of seven participants used health-related mobile apps, but their use was limited to making doctors' appointments, and they found it difficult to download and use health apps.

**Overall perception of the prototype:** The participants were positive about the prototype and were satisfied with its overall usability, regardless of their age. The survey results also revealed that users in their 20s, 40s, and 60s were highly satisfied, with an average satisfaction rating of 6 or higher for all items on a 7-point Likert scale. Furthermore, the final prototype improved user satisfaction compared with the portrait and landscape types with which users were highly satisfied in the pilot study (Figure 6). When the respondents were asked about their willingness to use the service if it was to be released in the manner in which the prototype was designed, they expressed a high level of willingness, regardless of their age. Despite being less prone to illnesses, the participants in their 20s were more likely to use the service and would recommend it to others. People in their 40s to 60s, who are more active in managing their health, often do not use disease management apps because they find it difficult to learn how to use new apps, but all 10 participants of this age group expressed interest in using the prototype. In addition to actively participating in the management of the disease under treatment, some of these participants stated that they would use the prototype to become aware of their physical condition through information in their medical records.

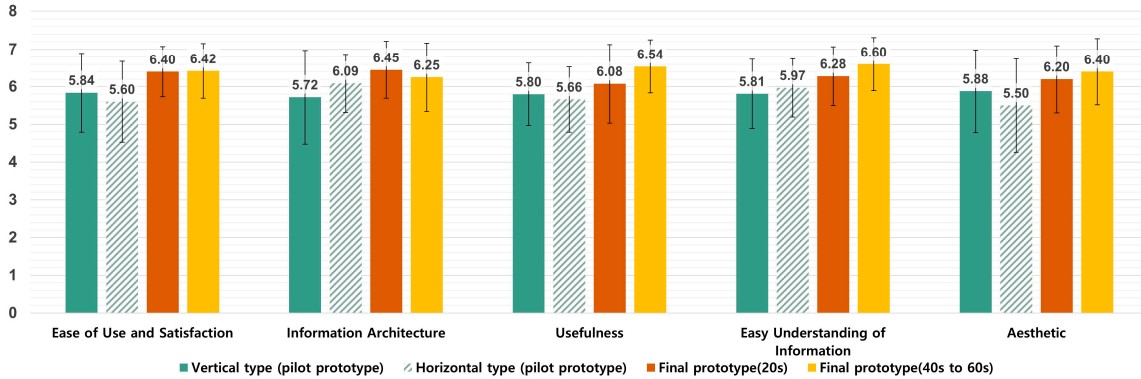

**Figure 6.** Graph showing user survey results.

**Evaluation of patient-centred PJM information delivery methods:** The participants were willing to use the integrated treatment information that was organised according to the PJM, as they stated that it was necessary. The greatest benefit was the ability to view previous medical treatments at a glance. Personal disease treatment data have previously been organised by medical institutions or in chronological order, which makes it difficult to check by disease. However, the app categorises treatment information by disease and allows patients to view their disease history quickly and conveniently. Furthermore, participants in their 40s and 60s cited the ability to recall what they were told about at the hospital as a significant advantage; they stated that it was very convenient and necessary to manage disease data in an integrated manner. As this age group is highly interested in disease management, there were many opinions stating that it is useful for users to be the subjects of their medical data, such as by being able to occasionally view their test records on the app.

*'The best part is that I can see my records at a glance, especially in situations where I need to see my previous treatment history or check my medication history at the hospital'. (P1-20s)*

*'I can see my health at a glance, so I think I can use it well'. (P11, 40s)*

However, some participants questioned whether the technology could aggregate and manage the data.

*'It's so good (to see it by disease), but I wonder if it's possible to aggregate and show the records like this'. (P12, 40s)*

*'I don't know if it's actually a feasible technology'. (P13, 40s)*

**Evaluation of information visualisation:** All participants, regardless of their age, agreed that the presentation of the prototype was appropriate for the mobile environment and did not feel unfamiliar. They also reported that the task/information was easy to understand and presented in a user-centred manner. All participants agreed that there was a clear distinction among the diseases through the use of colour coding; however, some participants felt that there was no semantic connection between the colours and diseases.

*'I think colour helps, like when you use colour to distinguish subway lines, but I don't know if it helps much'. (P1-20s)*

*'The different colours made it easier to distinguish, and I don't think people would know what the disease colours mean, so I think it's just right now'. (P3, 20s)*

The participants also found it intuitive to access a detailed screen from the card-type home screen. They felt that the detailed screen of the disease was organised vertically so that the eye could follow it naturally and the desired information was displayed at the bottom. However, some subjects in their 40s to 60s felt that it was difficult to select various types of information because the information selection button was in an on/off form; however, as the information could be selected and viewed, there was a smaller scrolling burden owing to the amount of information.

**Ideas for better ways to present information:** We obtained ideas for improving the prototype in terms of the information visualisation. First, several participants commented on the need to categorise recovered diseases. At present, the prototype is organised in chronological order of disease onset, which may be less important for recovered diseases; it would be more useful to view them separately or according to disease stage.

*'If there are multiple disease cards that have been cured, it would be nice to be able to see only the cured ones (cured ones and those under treatment at once)'. (P15, 40~60s)*

*'I don't think it's necessary to see a cured disease from the top'. (P5, 20s)*

Other comments included the need for a button to reset all selections when selecting the desired information on the detail screen and the need for a keyword search feature to make it easier to find information when the amount of information becomes overwhelming.

Most users found the language easy to understand, but some participants commented that some of the terms were unfamiliar; therefore, it would be preferable to have a separate explanation for medical jargon that is unfamiliar to non-medical users.

> 'When I look at what the doctors write, I see how it's expressed in English and how severe it is, but when I look at the diagnosis, I think it's important to change it to make it easier for us to see'. (P1, 20s)

> 'Most of the words themselves are written in an easy way, but since it is medical information, I think there are situations where difficult terms appear. In such cases, it would be good to have additional explanations'. (P6, 20s)

**User needs for additional information and features:** Users across all age groups wanted to know more about their healthcare, particularly regarding medications. Participants in their 20s, who were less experienced with medical treatments, also wished to view a record and information about the medications they were taking, whereas those in their 40s to 60s wished to view information regarding drug interactions, side effects, and misuse, as well as to have the ability to set alarms to help them to remember to take their medications during treatment.

The participants expressed that they would like to see features that make it easier for them to become more involved in their care, such as the ability to add their own information, reminders of doctors' appointments and treatments, integration with their prescriptions, and communication channels with their doctors.

In addition to information relating directly to their illness, they wished to access personalised information to help them to manage their health on a day-to-day basis, including basic health profile information such as height and weight, preventative measures against illness, and exercise and food recommendations.

## 6. Discussion

### 6.1. Importance of PJMs from a Patient's Perspective

In this study, we defined a patient-centred PJM, developed an information delivery method for effectively providing medical treatment information to patients, and fabricated and evaluated a mobile prototype. It appears that the medical/health data delivery method based on the proposed patient-centred PJM helped patients to understand their scattered treatment information. In particular, aggregation of information centred on individual diseases and the provision of information to users according to the treatment stage can help users to check the history of their diseases and participate actively in their healthcare.

### 6.2. Important Elements of the PJM UX

The results of this study indicate that users prefer more detailed information regarding their health and highlight the importance of designing a UX that makes health information easy to understand, enabling mobile health app searching.

In addition to organising information in a patient-centric manner, we used various visualisation techniques, which can be improved further in the future. First, the prototype used colours to differentiate between disease types, and this colour representation is familiar to most users. However, as colours may have certain meanings on their own (e.g., the colour red may be perceived as more serious or urgent than green), we need to be more careful when applying colours beyond the purpose of distinguishing diseases.

Language is also important for conveying information to users more easily. Thus, we attempted to use user-friendly words to explain health data, although medical terms may be difficult to understand. This drawback can be mitigated using a UX that provides a detailed explanation of any term upon tapping. Users have been highly satisfied with this feature in previous studies [55].

It is challenging to convey all of the detailed health information that patients wish to view in the form of visualisations because some information may be very specialised or, depending on the health condition of the patient, may need to be provided by a medical

professional. For example, patients with multiple diseases may wish to know how the different medications they are taking affect one another or may wish to know more about the medication. In such cases, it is ideal to provide a channel for them to directly contact a healthcare professional. Some healthcare apps offer a feature for contacting medical professionals [25]. It is possible to provide more personalised, professional, and detailed information to patients and improve their communication with healthcare professionals using such a feature and information from PHRs.

### 6.3. Limitations and Further Research

In this study, user evaluations were conducted with typical users in their 20s, 40s, and 60s. However, the proposed prototype will be used by users with various diseases and disabilities. Therefore, it is necessary to conduct a study that considers accessibility for such users and not simply their age. For example, the interview results highlighted the necessity of considering font sizes for elderly patients.

PHRs contain a variety of information, including public health data, hospital medical data, and personal health data; however, this study focused on disease information, which is classified as hospital medical data, for which a UX was proposed. Disease information is a part of personal health information and may not be useful for disease-free people. Such information must be linked with other health information to ensure better healthcare services. We found that the participants wanted to use the method and information provided in the prototype as the starting point to connect with various health management functions (e.g., dietary guidance based on the disease, a check-up schedule, exercise guidance, and preventive measures). Therefore, it is possible to develop a UX that can connect to other personal health data.

Furthermore, it is challenging to classify the stage of a disease because this requires a combination of various disease features, judgment by medical staff, and easy recognition by patients. In this study, we attempted to classify the stages of diseases and applied the classifications to PJMs based on the example of medical staff from various fields and users of various age groups (20s, 40s, and 60s). Consequently, we obtained the following stages: at the onset, before treatment, during treatment, after treatment, and full recovery. These results will be useful for the development of similar apps in the future. However, as many diseases and various treatment methods exist, it is necessary to develop a patient-centred PJM for more medical staff in the future. Moreover, as noted by the participants in this study, standardisation of information is necessary in implementations of mobile health services based on patient-centred PJMs, along with technical support. Specifically, efforts are required to define detailed items for collection through the platform by the type of health information, as well as to perform data standardisation to integrate various types of health information provided by different medical organisations using a person-centred approach, thus making them interchangeable. The classification of disease stages will also form part of the data standardisation process.

### 6.4. Contribution

This study contributes to the following four aspects. First, a new design for the PJM-based visualisation method was proposed for mobile apps, and the design was verified through user tests. As mentioned in the Introduction, few previous studies have considered patients or investigated the visual design of mobile apps in the case of PJMs. The design form shown in Figure 5 is meaningful in that it presents a new design proposal for a mobile app that considers the main timeline and primary users (i.e., patients), and it can be used as a cornerstone for related research. Second, a design guideline for health data visualisation is proposed through a literature review. Few visualisation studies have targeted health data, and most works have simply demonstrated the process of attempting to provide health data visualisation. The proposed guidelines in Table 4 can be used in various health data visualisation processes in the future. Third, a new methodology for usability evaluation of health research is proposed. It is challenging to conduct user evaluations in the prototype

process for health-related research because each user has different health data. Therefore, in this study, an experiment was conducted using the persona technique that is used in user-centred design, and the persona was verified by four medical experts. Fourth, a patient-centred PJM (Figure 2) verified by medical experts is presented. The main timeline is a critical element in the PJM, and no previous study has explained how to classify this element into stages or which tasks should be included in each stage. Thus, the contents of the proposed patient-centred PJM can be applied to other studies in the future.

## 7. Conclusions

We developed a prototype for delivering PHR information on mobile devices according to patient-centred PJMs and conducted user evaluations with Korean participants. A patient-centred PJM was established, and a prototype for mobile devices was subsequently built by arranging the information of the patient who had undergone treatment at a medical institution and applying various visualisation techniques. Finally, the UX of the prototype was evaluated. The results demonstrated that users could understand the information obtained from medical institutions according to the treatment stage for different diseases using the information delivery method of the prototype; this will promote the active participation of users in their treatment and recovery.

To date, most PJMs have primarily been used as strategic tools to facilitate communication between healthcare providers and stakeholders and to develop better healthcare services. Notably, the patient is the principal user of the PJM; thus, in addition to its use as a medical information analysis tool, the PJM can be adopted to initiate improved communication between medical professionals and patients.

With the recent accelerated development of health information technology, mHealth services such as PHRs will continue to diversify and advance. The proposed information delivery method based on patient-centred PJMs is anticipated to improve communication between various mHealth services and users. The improved interactions among medical stakeholders will promote a better understanding of health information among users, drive the active participation of patients in health management, and contribute to improving the overall quality of health.

**Author Contributions:** Conceptualization: H.K.K., Y.S. and C.O.; methodology and data analysis: J.L., Y.C. and H.P.; prototype: J.L., Y.C. and Y.S.; data analysis: J.L., Y.C. and H.P.; writing—original draft preparation: B.L. and H.P.; writing—review and editing: H.K.K. All authors have read and agreed to the published version of the manuscript.

**Funding:** This work was supported by a National Research Foundation of Korea (NRF) grant funded by the Korea government (MSIT) (No. NRF-2021R1F1A1063155). This research was also supported by the MSIT (Ministry of Science and ICT), Korea, under the ICAN (ICT Challenge and Advanced Network of HRD) program (IITP-2022-RS-2022-00156215) supervised by the IITP (Institute of Information and Communications Technology Planning and Evaluation).

**Institutional Review Board Statement:** This study was approved by the IRB (7001546-202300331-HR(SB)-003-02) of Kwangwoon University.

**Informed Consent Statement:** Informed consent was obtained from all subjects involved in the study.

**Data Availability Statement:** The data presented in this study are available on request from the corresponding author.

**Conflicts of Interest:** The authors declare no conflict of interest.

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
