# Peer review of "Visualisation of Information Using Patient Journey Maps for a Mobile Health Application"

_applsci, doi:10.3390/app13106067_

Round 1

Reviewer 1 Report

I thank the authors for their work. It is a well written and valuable article. Especially for mHealth developers and medical staff as well as researchers focusing on medical information and patient use. I only have a few comments: 

1. The description of the literature study does not contain information about the search strategy and inclusion procedure. The authors seem to have included valuable information, yet the needed information to check this is missing. This also counts for the result sections of the literature review: how many titles, abstracts and full texts screened and based on what excluded? Because of expected word-count issues, this can partly be solved by adding the search strategy and selection criteria in an additional file.

2. Paragraph ‘User needs for additional information and features’ (Line 592-605) shows essential information. Especially the information on disease and drug interactions will be difficult to include in a visualization based on individual diagnoses. It would be of great value to further discuss this in the discussion section (for example 6.2); especially looking at the impact of this on a next level prototype.

3. Figure 4 and 6: It would improve the readability of figure 4 when the Y-axis would start at 3 or 4 and would show a range of the scale in the vertical line of the graph.

Reviewer 2 Report

The subject of the paper is located in an important and current topic area, digitalization in healthcare. Topics such as usability and user experience, and thus also acceptance by patients, should be a central aspect of the transformation, and so the thrust of the work is expressly to be welcomed. There is still a need for further research in this area.

The presented work deals with a smartphone app for a patient journey.

For a journal publication, the current state has some weaknesses. In the overall view, the degree of innovation does not seem sufficiently clear. This should be focused on more strongly throughout the entire article. A pure look at the result of the user interface does not reveal the need for research, since common elements in the UI are essentially used here.

Likewise, the literature work for a journal article turns out to be very small. I consider a more detailed classification in the state of the art of science and technology to be necessary. This starts with the introduction, but also includes design decisions and a discussion of the own results in comparison to the literature.

Some questions also arise about the methodological approach:
- User research for the persona: how was the data for the persona determined? How were the attributes considered determined? Wouldn't, for example, information on technology experience/technology affinity also be relevant?
- Why was the initial design conducted before the literature research? Wouldn't it make more sense to first gather the state of the art and then design and test your own variants, possibly also in comparison to approaches from the literature?
- The test persons of the preliminary study and half of the test persons of the main study do not fit the persona / target group.
- The statistical analysis is incomplete. Typical information about the standard deviation, statistical significance of the described differences, validity tests (especially because the questionnaire is not established), limitations of the significance are missing. Further data, such as affinity for technology and relevant previous knowledge, would also have been interesting.

Other rather more detailed notes:
- The healthcare system and the level of digitalization, as well as people's attitudes toward health apps, differ greatly between countries. Therefore, it should already be clear in the abstract, introduction and conclusions for which country or group of countries the results are valid.
- What procedure was used to conduct the literature search and what criteria were used to select the sources? As already mentioned, the inclusion of further sources would be desirable.
- 2.2 does not fit under the heading "Literature Review". In addition, many statements are quite vague and hardly substantiated.
- 2.3 addresses "User Experience" in the title, but usability and technology acceptance are discussed. However, the inclusion of typical UX factors would be helpful for the entire work.
- Fig 1: only process steps 4 and 5 are shown
- In Table 1: Q3 and Q4 are identical. In addition, the specification of the scale used is missing.
- Table 3: How were the tasks and questions defined?
- 5.3: The phases of prototype design and study implementation should not be mixed in the structure of the text.
- Finally, proofreading should be done again

Round 2

Reviewer 2 Report

Many thanks to the authors for the revised version and for taking my comments into account. The new text sections and the attached explanations have improved my understanding about the background of the paper.
In my opinion, the article has been significantly improved. The following comments remain:
In 2.3 I would appreciate if the UX questionnaires AttraktDiff, UEQ and meCue were mentioned in the discussion.
In 5.1.4, I think it should be mentioned more explicitly that the quantitative results are so close together that no statistically significant statements about the differences are possible.
The deviation of the test subjects from the target group is understandable and of course a problem known to all scientists. Nevertheless, it is a limitation of the significance and this should be mentioned accordingly.

Author Response

First, we would like to thank you for giving valuable and thoughtful reviews. Here, we present some possible improvements in response to the reviewers’ comments and also explain some points of miscommunication. The sentences in the italic font are the reviewer’s comments. We have added the sentences and references in the red color in the revised paper.

Q1. Many thanks to the authors for the revised version and for taking my comments into account. The new text sections and the attached explanations have improved my understanding about the background of the paper.
In my opinion, the article has been significantly improved. The following comments remain:
In 2.3 I would appreciate if the UX questionnaires AttraktDiff, UEQ and meCue were mentioned in the discussion.

--> Thank you for our valuable comments. We have added the following sentences in section 2.3 by stating “Traditionally, a common way to measure the UX of an interactive product is the AttrakDiff questionnaire, the User Experience Questionnaire (UEQ), or the modular evaluation of key Components of User Experience (meCUE)[26-29]. AttrakDiff assesses the user’s feelings about the cognitive artifact with the help of 28 pairs of opposite adjectives [26]. Users can evaluate both the perceived pragmatic quality, the hedonic quality, and the attractiveness. UEQ covers the user experience aspects: attractiveness, efficiency, perspicuity, dependability, stimulation, and novelty[27]. It allows the users to express feelings, impressions, and attitudes that arise when experiencing the product under investigation in a very simple and immediate way. meCUE was developed to measure key components of UX in a comprehensive and unified way. meCUE is a questionnaire with 34 items covering the components: product perceptions (usefulness, usability, visual aesthetics, status, commitment), user emotions (positive, negative), consequences of use (intention to use, product loyalty), and overall evaluation [28]. However, these methods are not specialized for mobile healthcare services, but are aimed at general interactive systems.”

Added references are:

  1. Hassenzahl, M.; Burmester, M.; Koller, F. AttrakDiff: Ein Fragebogen zur Messung wahrgenommener hedonischer und pragmatischer Qualität. In Mensch & Computer 2003: Interaktion in Bewegung; B.
  2. G. Teubner: Stuttgart, Germany, 2003; pp. 187–196. 16. Laugwitz, B.; Held, T.; Schrepp, M. Construction and Evaluation of a User Experience Questionnaire. Comput. Sci. 2008, doi:10.1007/978-3-540-89350-9_6.
  3. Minge, M.; Riedel, L. meCUE – Ein modularer Fragebogen zur Erfassung des Nutzungserlebens [meCUE - A modular questionnaire for capturing the user experience]. Mensch und Comput. 2013, 9, 89–98.
  4. Díaz-Oreiro, I.; López, G.; Quesada, L.; Guerrero, L.A. Standardized Questionnaires for User Experience Evaluation: A Systematic Literature Review. Proceedings 2019, 31, 1014.

Q2: In 5.1.4, I think it should be mentioned more explicitly that the quantitative results are so close together that no statistically significant statements about the differences are possible.
The deviation of the test subjects from the target group is understandable and of course a problem known to all scientists. Nevertheless, it is a limitation of the significance and this should be mentioned accordingly.

--> The authors agree the more explicit explanation is needed for the quantitative results. We have added the following sentence in section 5.1.4 by stating “However, these results were not statistically significant because there were only a small number of participants, and the scores for each prototype were slightly different.”